# Mining Transcriptomic Data to Uncover the Association between CBX Family Members and Cancer Stemness

**DOI:** 10.3390/ijms232113083

**Published:** 2022-10-28

**Authors:** Patrycja Czerwinska, Andrzej Adam Mackiewicz

**Affiliations:** 1Department of Cancer Immunology, Poznan University of Medical Sciences, 61-866 Poznan, Poland; 2Department of Diagnostics and Cancer Immunology, Greater Poland Cancer Centre, 61-866 Poznan, Poland

**Keywords:** CBX, HP1, TRIM28, Polycomb, cancer stemness, mRNA-SI, TCGA

## Abstract

Genetic and epigenetic changes might facilitate the acquisition of stem cell-like phenotypes of tumors, resulting in worse patients outcome. Although the role of chromobox (CBX) domain proteins, a family of epigenetic factors that recognize specific histone marks, in the pathogenesis of several tumor types is well documented, little is known about their association with cancer stemness. Here, we have characterized the relationship between the CBX family members’ expression and cancer stemness in liver, lung, pancreatic, and uterine tumors using publicly available TCGA and GEO databases and harnessing several bioinformatic tools (i.e., Oncomine, GEPIA2, TISIDB, GSCA, UALCAN, R2 platform, Enrichr, GSEA). We demonstrated that significant upregulation of CBX3 and downregulation of CBX7 are consistently associated with enriched cancer stem-cell-like phenotype across distinct tumor types. High CBX3 expression is observed in higher-grade tumors that exhibit stem cell-like traits, and CBX3-associated gene expression profiles are robustly enriched with stemness markers and targets for c-Myc transcription factor regardless of the tumor type. Similar to high-stemness tumors, CBX3-overexpressing cancers manifest a higher mutation load. On the other hand, higher-grade tumors are characterized by the significant downregulation of CBX7, and CBX7-associated gene expression profiles are significantly depleted with stem cell markers. In contrast to high-stemness tumors, cancer with CBX7 upregulation exhibit a lower mutation burden. Our results clearly demonstrate yet unrecognized association of high CBX3 and low CBX7 expression with cancer stem cell-like phenotype of solid tumors.

## 1. Introduction

Decades of research have aimed to characterize the mechanisms supporting a specific population of cancer cells termed cancer stem cells (CSCs). These cells evade the process of differentiation during tumor growth, preserving the self-renewal capacity that further accelerate the expansion of both stem and non-stem cancer cells [1,2]. The stem-like compartment of the tumor seems to be less susceptible to standard therapies and to play a vital role in the process of metastatic spreading and tumor relapse. A growing body of evidence indicates that stem cell-associated molecular features, collectively known as cancer stemness, are biologically relevant in the development and progression of tumor [3]. Recently, these features were recognized as valuable predictive or prognostic characteristics that ultimately revealed a strong positive correlation between a dedifferentiated tumor phenotype and worse patient outcomes across many types of cancer [4,5,6]. Therefore, specific gene expression signatures that allow for grading cancer stemness constitute an indisputable step in the development of novel therapeutic regimens that would eradicate cancer stem cells.

Recent studies suggest that non-stem cancer cells might acquire a stem-like state without any genetic manipulation [7]. Here, the epigenetic alterations come to the light, especially considering the importance of epigenetic regulators in normal stem cell maintenance. Both chromatin compaction and gene expression are tightly regulated by epigenetic modifications, including DNA methylation or post-translational modifications of histone proteins [8]. Specific chemical groups (i.e., methyl or acetyl groups) are deposited on histone proteins by distinct enzymes, known as “writers”, and subsequently recognized by effector proteins, known as “readers” (or, when needed, removed by other enzymes—“erasers”), profoundly affecting gene expression [9]. Among others, the chromobox (CBX) domain family members—named after the N-terminally encoded Chromatin Organization Modifier (Chromo) domain—function as epigenetic readers that control gene expression [10] and are known from their involvement in various cancers, including breast [11], colon [12], and lung [13] tumors. To date, there are eight CBX family members, categorized into two distinct subgroups—based on their structural differences and functions—the heterochromatin protein 1 (HP1) group and polycomb protein (Pc) group [14].

Cbx1, Cbx3, and Cbx5, also known as HP1β, HP1γ, and HP1α, respectively, recognize methylated histone H3K9, an established marker of the transcriptionally repressed state of chromatin, facilitating heterochromatin maintenance [15]. HP1 members interact through their C-terminal chromoshadow (CSD) domain with a Tripartite Motif 28 (Trim28) scaffold protein, that together with other chromatin modifying factors, is recruited to the specific genomic loci by Krüppel-Associated Box Zinc Finger (KRAB-ZNF) transcription factors. HP1 proteins stabilize Trim28-containing complexes, leading to silencing of euchromatic genes by KRAB-ZNF TFs [16,17].

Cbx2, Cbx4, Cbx6, Cbx7, and Cbx8 recognize trimethylated histone H2K27 and constitute the canonical components of polycomb repressive complex 1 (PRC1), a histone ubiquitin ligase that requires H3K27me3 for chromatin occupancy [18]. H3K27me3 is deposited by PRC2, and its’ methyltransferase activity is dependent on three subunits: embryonic ectoderm development (EED), suppressor of zeste 12 (SUZ12) and either enhancer of zeste homologue 2 (EZH2) or its paralogue EZH1. The catalytic activity of PRC1 is mediated by the protein really interesting new gene 1A (RING1A) or its paralogue RING1B, and function in concert with PRC2 to epigenetically silence target genes to maintain transcriptional programs and ensure cellular identity [19].

It is well known that PRC1 and PRC2 complexes regulate stem cell pluripotency, cell fate decisions, and development and are dysregulated in many cancer types, facilitating cancer stem cell maintenance [20,21]. Additionally, Trim28-based gene repression machinery has a grounded place in the stem cell regulatory mechanisms [22,23], and recent reports demonstrate the involvement in cancer development and progression, especially by supporting the cancer stem cell population [24,25,26]. Several studies have investigated the role of Trim28- or PRC1-interacting CBX family members in cancer development and progression. However, little is known about CBX association with cancer stemness.

Here, we have exploited publicly available transcriptomic data from 4 distinct types of solid tumors, namely liver, lung, pancreatic, and uterine carcinoma, to delineate the association between CBX family members and cancer stemness measured with previously reported stemness scores [3,4,5,6]. Firstly, we analyzed the expression pattern of all CBX family members in tumor vs normal adjacent tissues using publicly available TCGA [27], Oncomine [28] and CPTAC data [29] from the cBioportal, Oncomine, and UALCAN databases, respectively. Using the GSCA platform, we determined the frequency of alterations in CBX family members. According to the GEPIA2 database [30], for several CBX members, we reported a significant association with TCGA cancer patients’ survival, mostly with a worse outcome. Furthermore, the correlation with clinicopathologic features revealed significant upregulation of CBX3 followed by significant downregulation of CBX7 in more advanced disease and in higher-grade tumors, as determined with the TISIDB database.

Using a transcriptome-based stemness index (mRNA-SI) and other stem cell-derived gene expression signatures [3,4,5,6], we analyzed the association between CBX family members and cancer dedifferentiation status. We observed that regardless of the TCGA tumor type or the stemness score, the expression of CBX3 is significantly positively and the level of CBX7 is significantly negatively associated with cancer dedifferentiation, which was further validated with additional datasets from the R2 database.

Furthermore, gene set enrichment analysis (GSEA) [31] revealed that transcriptome profiles of CBX3^HIGH^ tumors are significantly enriched, while CBX7^HIGH^ tumors are significantly depleted with stemness markers, respectively, further supporting our first observation. We further confirmed a robust enrichment of CBX3-associated transcriptome profiles with several hallmarks of cancer previously identified as upregulated in stemness-high tumors [3]. The opposite results were observed in CBX7^HIGH^ tumors. Moreover, as determined with the Enrichr analysis tool, the CBX3-associated transcription factor protein–protein interaction network is significantly enriched with c-Myc and Oct-3/4 transcription factors, which are known pluripotency markers.

Our results are the first that determine the association between CBX family members and cancer stemness in several distinct types of solid tumors using transcriptomic data from TCGA and R2 databases. Our data clearly demonstrate yet-unrecognized association of high CBX3 expression with cancer stem cell-like phenotype of solid tumors, regardless of the tumor type. On the other hand, this is the first report of negative association between CBX7 upregulation and cancer stemness. Both proteins are strictly associated with the regulation of gene expression, although harnessing distinct modes of action. Further studies are needed to determine the molecular mechanism of cancer stem cell-like state acquisition in CBX3^HIGH^ and CBX7^LOW^ tumors.

## 2. Results

### 2.1. CBX Family Members Are Differentially Expressed in Solid Tumors and Possess Distinct Prognostic Values

Firstly, we analyzed the expression of eight CBX family members, namely CBX1/HP1β, CBX2, CBX3/HP1γ, CBX4, CBX5/HP1α, CBX6, CBX7, and CBX8, in selected solid tumors and relevant normal tissues using the TCGA data for liver hepatocellular carcinoma (LIHC), lung adenocarcinoma (LUAD), pancreatic adenocarcinoma (PAAD), and uterine endometrial carcinoma (UCEC). The expression of CBX genes differs across solid tumors (Appendix A) and for CBX1, CBX2, CBX3, CBX4, CBX5, and CBX8 is predominantly upregulated, while for CBX6 and CBX7 is mostly downregulated in tumor tissues (Figure 1A–G). This was further confirmed with additional datasets for lung, liver and pancreatic tumors from the Oncomine database (Appendix A). Specifically, we observed significant upregulation of CBX1, CBX3, and CBX5 members, followed by robust downregulation of CBX7 in tumor tissues. Moreover, we analyzed the protein level of distinct CBX members using CPTAC data for LIHC, LUAD, PAAD, and UCEC tumors and further validated our first observation (Appendix A). A predominant positive correlation between CBX members in tumor tissues—except for CBX7—stays in line with abovementioned results (Figure 1H).

Next, we looked at the frequency of alterations in CBX family members in TCGA data. Using the GSCA platform [32], we observed that across 4 tested tumor types (1641 profiled samples), the frequencies of single nucleotide variations (SNV) in CBX encoding genes were relatively low (Appendix A) and ranged from 0.61% to 1.46% of all tested samples for CBX7 and CBX4, respectively. The SNVs were predominantly observed in UCEC tumors and missense mutations were the most frequent type of alteration (Appendix A). We also observed that several CBX members are frequently amplified in cancer tissues, especially CBX1, CBX2, CBX3, CBX4, and CBX8 in LUAD and LIHC tumors. On the other hand, CBX6 and CBX7 are deleted, particularly in LUAD and PAAD (Appendix A).

Furthermore, we analyzed the prognostic values of specific CBX members in selected solid tumors. We observed that higher expression of CBX1, CBX2, CBX3, CBX5, CBX6, and CBX8 is significantly associated with a worse overall survival of LIHC patients (Figure 2A). As for LUAD patients, we observed significant association of CBX3 and CBX5 upregulation with a worse survival. CBX3 upregulation is also related to a shorter survival of PAAD patients. On the other hand, the upregulation of CBX2, CBX6, CBX7, and CBX8 significantly correlates with a better overall survival of these patients (Figure 2A). As for relapse-free survival, we observed significant association of high CBX1, CBX2, and CBX3 expression with a worse RFS survival of LIHC patients (Figure 2B). Moreover, CBX3 upregulation significantly correlates with a shorter relapse-free survival of LUAD and PAAD patients. On the other hand, higher levels of CBX6, CBX7, and CBX8 associate with a better relapse-free survival of PAAD patients.

### 2.2. The Expression of Specific CBX Members Correlates with Tumor Stage and Tumor Grade

We further analyzed the level of CBX family members in relation to the tumor stage and observed significantly higher CBX1, CBX2, CBX3, and CBX4 expression in more advanced LIHC tumors (Figure 3A), higher CBX2, CBX3, and CBX5 and lower CBX7 levels in higher stage LUAD tumors (Figure 3B), lower CBX1, CBX5, CBX6, and CBX7 expression in more advanced PAAD tumors (Figure 3C), and higher CBX1, CBX3, CBX5, and CBX6 levels in higher stage UCEC tumors (Figure 3D).

Also, we analyzed the expression of CBX members in regard to tumor dedifferentiation status (histological tumor grade). As presented in Figure 4, we observed significant positive correlation between a high CBX1, CBX2, CBX3, CBX5, and CBX8 expression and higher tumor grade in LIHC and UCEC tumors. On the other hand, we observed significant negative association between a high CBX7 expression and higher tumor grade in LIHC and PAAD tumors.

### 2.3. CBX3 Is Positively While CBX7 Is Negatively Associated with Tumor Stemness

As previously reported, solid tumors exhibit different levels of stemness that can be quantified with specific molecular signatures (Appendix A). In our studies, we have implemented the mRNA-based stemness index (mRNA-SI) previously defined by Malta T. et al. [3] using the machine learning algorithm. Moreover, we used the “BenPorath ES core nine” gene signature [4], as histologically poorly differentiated tumors show preferential overexpression of genes normally enriched in embryonic stem cells (ESC). The reactivation of the ESC-like program in cancer strongly predicts metastatic potential and patients’ death, therefore, we also used “Bhattacharya ESC” and “Wong ESC core” gene signatures to quantify tumor stemness in selected TCGA solid tumors [5,6].

We observed that in all 4 tested tumor types and using several distinct stemness quantifiers, the expression of CBX3 is significantly positively, while the level of CBX7 is significantly negatively associated with cancer stemness (Figure 5A–D) in TCGA data. The expression of CBX1, CBX2, CBX5, and CBX8 significantly positively correlates with cancer stemness in LIHC, LUAD, and UCEC tumors, while CBX6—negatively associates with cancer stemness in LUAD and PAAD tumors. This was further validated with additional GEO datasets (Appendix A). As presented in Figure 5E–H the expression of CBX3 is significantly positively, and CBX7—negatively correlated with cancer stemness in all 4 tested tumor types. As for CBX1 and CBX2—we observed positive association with cancer stemness in liver, lung and endometrial tumors, while CBX4, CBX5, CBX6, and CBX8 exhibit rather weak correlation with cancer stemness in a tumor-specific manner. Therefore, we have further focused on CBX3 and CBX7 proteins, as their association with cancer stemness is most prominent and universal regardless of the tumor type.

Using the GSEA, we compared the CBX3 and CBX7-associated transcriptome profiles with a priori defined stemness-associated gene signatures derived from the StemChecker database (Appendix A) [33]. Appendix A presents CBX3 and CBX7-associated transcriptome profiles in LIHC, LUAD, PAAD, and UCEC tumors defined as all significantly correlated genes (Spearman correlation, FDR < 0.01) to CBX3 and CBX7, respectively. We confirmed significant enrichment of CBX3-associated transcriptome profiles (Figure 6A–D) followed by significant depletion of CBX7-related transcriptome profiles (Figure 6E–H) with stem cell markers in all 4 tested tumors, which was further validated with several additional GEO datasets (Figure 6I–L). This strongly support our first observation of CBX3 being positively and CBX7 being negatively correlated with tumor stemness.

### 2.4. The “Hallmark of Cancer” Terms in CBX3 and CBX7-Associated Gene Expression Profiles

Previously, Malta et al. [3] has demonstrated a significant enrichment of several hallmark of cancer terms in the cancer stemness mRNA-SI gene signature. Especially, the targets for c-Myc and E2F transcription factors are over-represented in mRNA-SI (Figure 7A,B). Here, we verified whether the CBX3- and CBX7-associated gene expression profiles reflects the enrichment with specific hallmarks of cancer observed in mRNA-SI. As presented in Figure 7C, the CBX3-associated transcriptome profiles are significantly enriched with the targets for c-Myc and E2F transcription factors and genes involved in mTOR signaling pathway or G2/M cell cycle checkpoint in all 4 tested tumors, that strongly mirrors the results for the mRNA-SI gene signature. On the other hand, CBX7-associated transcriptome profiles are significantly depleted with the targets for c-Myc or E2F transcription factors, as well as the markers of mTOR signaling or G2/M checkpoint, regardless of the tumor type (Figure 7D). Similar results were obtained for additional GEO datasets (Appendix A).

We further validated this observation with the Enrichr [34], an integrative web-based enrichment analysis tool, that uses the ARCHS4 web resource to define significantly co-expressed genes with the gene of interest in a huge collection of RNA-Seq data from large projects, such as GTEx or TCGA. Among the pathways significantly enriched in CBX3-associated gene expression profiles, the targets for c-Myc and E2F transcription factors followed by the markers of mTOR signaling pathway and G2/M checkpoint are mostly prominent (Figure 7E). On the other hand, we did not observe the enrichment of the abovementioned terms in CBX7-associated gene expression profiles.

Moreover, the Enrichr analysis of transcription factor target genes within a tested gene set revealed a significant enrichment of CBX3-associated gene expression profile with targets for Oct-3/4 (encoded by POU5F1)—a key regulator of cell pluripotency and self-renewal (Figure 7F). This further supports our first observation of positive association between CBX3 and cancer stemness.

### 2.5. CBX3 Overexpression and CBX7 Downregulation Associate with the Abundance of Genomic Alterations

We further analyzed the abundance of genomic alterations in stem cell-like tumors. As presented in Figure 8A, we observed significant positive correlation between mRNA-SI and genomic alterations in LIHC, LUAD, PAAD, and UCEC tumors as determined with several distinct aberration quantifiers [35]—silent and non-silent mutations rates, the aneuploidy score, fraction of altered genome, and homologous recombination defect score. In LIHC, LUAD, and PAAD we also observed significant positive association between CBX3 level and the abundance of genomic alterations (Figure 8B), that strongly mirrors the results obtained for mRNA-SI. On the other hand, CBX7 expression negatively correlates with genomic alterations, especially in LUAD and PAAD tumors (Figure 8B), which corresponds to the negative association between CBX7 and cancer stemness.

## 3. Discussion

To date, the relation between CBX family members and cancer stemness was scarcely investigated. Here, we have harnessed the transcriptomic data from TCGA, Oncomine and R2 databases and proteomic data from CPTAC database for liver, lung, pancreatic, and uterine carcinomas and demonstrated that: (i) CBX family members are predominantly overexpressed (except for CBX6 and CBX7) and scarcely mutated in tumor tissues; (ii) higher levels of CBX3 associate with worse overall and relapse-free survival in LIHC, LUAD, and PAAD tumors, while for other CBX members, the association is tumor specific; (iii) for most CBX family members (except CBX7), the upregulation associates with more advanced disease and with higher-grade tumors; (iv) CBX3 level significantly positively, while CBX7—negatively correlates with cancer stemness scores, respectively, regardless of the tumor type or the stemness quantifier; (v) the transcriptome profiles of CBX3^HIGH^ expressing solid tumors are significantly enriched, while CBX7-associated transcriptome profiles are significantly depleted with stem cell markers; (vi) the enrichment of CBX3-associated transcriptome profiles with targets for c-Myc and E2F transcription factors considerably reflects the enrichment determined for tumors exhibiting high mRNA-SI levels, (vii) and high CBX3 expressing tumors manifest elevated genomic aberration frequencies which corresponds to higher mutation burden in cancer stem cell-like tumors.

Both groups of CBX proteins are involved in the regulation of heterochromatin, gene expression, and developmental processes, albeit they use distinct binding partners and recognize different post-translational modifications deposited on histone proteins [21].

To mediate their functions, Cbx1, Cbx3, and Cbx5—members of the HP1 group—specifically interact with di-/trimethylated lysine 9 on the histone H3, homo- or heterodimerize, and further form multimolecular complex with transcriptional corepressor Trim28, histone methyltransferase SETDB1 and NuRD-histone deacetylase complex [36]. Despite their structural similarity, the HP1 group proteins have some distinct, non-redundant functions and localization patterns [37]. Specifically, Cbx5/HP1α is present mainly in heterochromatic regions, Cbx1/HP1β is found in both hetero- and euchromatic regions and Cbx3/HP1γ is primarily located in euchromatin, within the transcribed regions of active genes [38]. Here, it regulates transcriptional elongation and co-transcriptional mRNA processing [39,40]. During cell differentiation, the localization of HP1 proteins is submitted to robust changes which accounts for the chromatin organization. The HP1 proteins are bone fide transcriptional repressors. However, Cbx3/HP1γ is also associated with the transcriptional activation of direct target genes and unlike other HP1 isoforms, Cbx3/HP1γ may sustain gene expression [39,41]. Generally, HP1 binds to specific regulatory DNA sequences to promote local compaction of genomic regions and subsequently block the commitment of pluripotent cells toward differentiation.

In our data, the most prominent and consistent positive association with cancer stemness was observed for CBX3. We also revealed a significant positive correlation of CBX1 and CBX5 with tumor dedifferentiation status, although not in all tested tumor types. CBX3 overexpression was previously shown to promote tumor progression and to predict worse survival of lung [13,42], gastric [43], pancreatic, breast [11], glioma [44], osteosarcoma [45], liver [46], and kidney tumors [47]. CBX3 is significantly upregulated in osteosarcoma tumor stem cells and facilitates their self-renewal [48]. Also, He Z. et al. [49] have reported that esophageal squamous cell carcinoma (ESCC) acquire stem cell-like characteristics through CBX3-dependent inactivation of p53/p21 pathway.

Here, we show for the first time that high CBX3 expression associates with cancer dedifferentiation status and CBX3^HIGH^ tumors exhibit the cancer stem cell-like phenotype. Moreover, CBX3 overexpressing tumors manifest higher frequencies of genomic alterations, which strongly reflects the elevated mutation burden of cancer stem cell-like tumors. Mutation accumulation during cancer development and progression might result in proliferative and survival advantage of selected cell clones, that acquire self-renewal potential and therapy resistance [50]. Cancer stemness is positively associated with mutation load [51], and CBX3^HIGH^ tumors display elevated abundance of genomic alterations, further supporting our first observation.

To date, Cbx3/ HP1γ was known to maintain the balance between stem cell identity and differentiation [37]. Cbx3/HP1γ suppresses spontaneous differentiation and facilitates self-renewal of normal stem cells by repressing the genes that promote cell differentiation. Our data indicate that CBX3 also facilitates cancer stemness, although direct mechanism remains elusive.

HP1 proteins are indispensable for the stability of Trim28-containing complexes that mediate gene repression in specific genomic loci (recognized by KRAB-ZNF transcription factors) [16,17]. Recently, we have demonstrated that TRIM28 (also known as TIF1β) is highly expressed in higher grade tumors that exhibit stem cell-like traits [25,26]. In contrast to other TIF1 members, only TRIM28-associated gene expression profiles were robustly enriched with stemness markers regardless of the tumor type [52]. As Trim28—a scaffold protein that recruits to the chromatin several other epigenetic factors, altering the activity of underlying transcriptional mechanisms, requires HP1 proteins for stable transcriptional repression, we suggest that both Trim28 and Cbx3/HP1γ are needed for cancer stemness maintenance.

As for Pc (Polycomb) group of CBX proteins, Morey L. et al. [53] have previously reported their nonoverlapping functions in embryonic stem cell pluripotency and differentiation. They demonstrated that in proliferating ESC cells and during the differentiation process distinct Cbx proteins are incorporated into the PRC1 complexes that exert specific functions. Target specificity of PRC1 and PRC2 in the pluripotent state is dictated solely by Cbx7, and no other PRC1-associated Cbx members. Moreover, in pluripotent cells, Cbx7-containing PRC1 complexes mediate the repression of CBX2, CBX4, and CBX8 genes to prevent premature differentiation. On the other hand, during lineage specification, the expression of CBX7 and pluripotency genes is tuned down by PRC1/PRC2. In cancer, Cbx7 was shown to possess dualistic role, either promoting or suppressing cancer progression, depending on cancer type and molecular interactors [54]. Cbx7 acts as tumor inhibitor and is significantly downregulated in bladder [55], breast [56], liver [57], colon [58], thyroid [59], glioma [60], and pancreatic cancers [61]. Also, CBX7 upregulation correlates with a better survival of tumor patients [54]. On the other hand, Cbx7 acts as an oncogene in gastric tumors by suppressing p16/CDKN2A and supporting AKT1 signaling pathway [62,63]. CBX7 is associated with poor prognosis in ovarian clear cell adenocarcinoma [64] by modulating the level of the tumor necrosis factor-related apoptosis-inducing ligand (TRAIL) and inhibiting apoptosis. Therefore, the expression and effects mediated by CBX7 are tissue and cell type specific, and exert distinct molecular mechanisms.

Our results are in line with previously reported inhibitory role for Cbx7 in tumorigenesis. High CBX7 expression significantly associates with cancer stemness attenuation and CBX7^HIGH^ expressing tumors are robustly depleted with markers of stem cells, regardless of the tested tumor type. Moreover, CBX7^HIGH^ tumors exhibit lower mutation load, which is the opposite for cancer stem-cell like phenotype. We suggest that in contrast to normal stem cells, Cbx7 represses cancer stemness. Recently, Huang Z. et al. [55] have demonstrated that CBX7 upregulation significantly affects cancer stemness in bladder cancer. CBX7 depletion promoted cancer cell aggressiveness, and associated with a significant increase in the expression of cancer stem cell markers. Also, CBX7 knockdown in breast cancer cells increased the frequency of cancer stem cell-like population and reinforced in vitro self-renewal and in vivo tumor-initiating ability of those cells [65]. On the other hand, Cbx7 positively regulates the stem cell-like properties of gastric cancer cells [63]. Therefore, the role of CBX7 in modulating cancer stemness might be affected by the tumor origin and the prevalence of distinct interacting proteins and should be considered in a tumor-specific manner.

Nowadays, it is well-established that the epigenetic dysregulation accelerates tumorigenesis, and at least partially, facilitates cancer stemness. Several reports suggest, that CBX family members possess the potential to become druggable targets in distinct tumor types, including breast, lung, ovarian, esophageal, and colon cancer [11,66,67,68,69]. However, these reports do not raise the question of therapeutic targeting of the stem cell-like phenotype of solid tumors and do not prove the engagement of CBX family members in molecular mechanism modulating cancer stemness. Albeit several molecular inhibitors of CBX proteins are already under the development [14,70,71], further studies are necessary to determine whether they provide advantages in the treatment of de-differentiated tumors that strongly exhibit cancer stem cell-like characteristics.

## 4. Materials and Methods

### 4.1. TCGA Data

In the current study, we used publicly available data for 4 solid TCGA tumors: liver hepatocellular carcinoma (LIHC), ling adenocarcinoma (LUAD), pancreatic adenocarcinoma (PAAD), and uterine corpus endometrial carcinoma (UCEC) from the cBioportal (www.cbioportal.org, accessed on 5 January 2022) [27] and the R2 Genomics Analysis and Visualization Platform (http://r2.amc.nl, accessed on 3 January 2022). All data is available online, and the access is unrestricted and does not require patients’ consent or other permissions. The use of the data does not violate the rights of any person or any institution.

### 4.2. Transcriptomic Data

The RNA sequencing–based mRNA expression data were directly downloaded from the cBioportal. RNASeq V2 from TCGA is processed and normalized using RSEM [72]. Specifically, the RNASeq V2 data in cBioPortal corresponds to the rsem.genes.normalized_results file from TCGA. The Spearman’s correlation was used for detection of co-expressed genes with *p*-value < 0.05 and FDR < 0.01 as cut-offs.

### 4.3. Oncomine Data

The data regarding CBX family members’ expression in additional datasets for liver, lung, pancreatic or endometrial cancers (Appendix A) was retrieved from the online database, Oncomine (https://www.oncomine.org/resource/login.html, accessed on 28 December 2021) [28]. Differences in mRNA expression between cancer and adjacent normal tissues were detected using the following threshold parameters: *p*-value < 0.05, |fold-change| > 1.5, and gene ranking in the top 10%.

### 4.4. Mutation Status

The mutation profile of CBX family members in LIHC, LUAD, PAAD, and UCEC tumors was analyzed using the Mutation panel of the GSCA database (http://bioinfo.life.hust.edu.cn/GSCA/#/, accessed on 28 December 2021) [32].

### 4.5. Proteomic Data

The UALCAN dataportal [73] was applied to determine the total protein levels of CBX family members in LIHC, LUAD, PAAD, and UCEC using data from Clinical Proteomic Tumor Analysis Consortium (CPTAC) Confirmatory/Discovery datasets [29].

### 4.6. Overall Survival and Relapse-Free Survival

The association between CBX family members expression and patients’ overall survival (OS) or relapse free-survival (RFS) was analyzed with the Survival_Map panel of the GEPIA2 database (http://gepia2.cancer-pku.cn/#index, accessed on 7 January 2022) [30]. The hazard ratio was estimated using the Mantel-Cox test using the median CBX family members’ expression as a cut-off.

### 4.7. Disease Stage and Histologic Tumor Grade

The association between CBX family members’ expression and the disease stage or the histologic tumor grade was assessed using the TISIDB portal (http://cis.hku.hk/TISIDB/index.php, accessed on 8 January 2022) [74]. The correlation was calculated using Spearman’s rank correlation coefficient (r).

### 4.8. Stemness-Associated Scores

The mRNAsi stemness score [3] and other stemness signatures (Ben-Porath_ES_core, Wong_ESC_core, Bhattacharya_ESC) used in this study to score the cancer stemness were previously described [4,5,6]. For the Gene Set Enrichment Analysis, additional stemness signatures (Appendix A) were downloaded from the StemChecker web server (http://stemchecker.sysbiolab.eu/, accessed on 10 January 2022) [33] or the Molecular Signatures Database (MSigDB, http://www.broad.mit.edu/gsea/.msigdb/msigdb_index.html, accessed on 10 January 2022) [75].

### 4.9. Gene Set Enrichment Analysis

The Gene Set Enrichment Analysis (GSEA, http://www.broad.mit.edu/gsea/index.html, accessed on 10 January 2022) [31] was used to detect the coordinated expression of a priori defined groups of genes within the tested samples. Gene sets are available at the MSigDB (http://www.broad.mit.edu/gsea/.msigdb/msigdb_index.html, accessed on 10 January 2022). All significantly correlated genes (previously ranked based on their Spearman correlation coefficient r value) were imported to GSEA. The GSEA was run according to the default parameters: each probe set was collapsed into a single gene vector (identified by its HUGO gene symbol), permutation number = 1000, and permutation type = “gene-sets.” The FDR (<0.01) was used to correct for multiple comparisons and gene set sizes.

### 4.10. Enrichr

Additionally, the GSEA results were further validated with the Enrichr (https://maayanlab.cloud/Enrichr/, accessed on 13 January 2022) [34]—an integrative web-based software application providing various types of computing gene set enrichment and visualization summaries of collective functions of single genes or gene lists. We used top 100 most relevant genes (identified with ARCHS4 RNA-seq gene-gene co-expression matrix) for a queried gene (CBX3 or CBX7) to determine significantly enriched pathways (MSigDB Hallmark 2020) or to detect potential targets for known transcription factors (Transcription Factor PPIs).

### 4.11. Validation Sets

Additional datasets used in this study (GSE15765, GSE2109, GSE43580, GSE21501, GSE184585, GSE36389; for further details see Appendix A) were obtained from the R2 Genomics Analysis and Visualization Platform (http://r2.amc.nl, accessed on 5 January 2022). All datasets were analyzed online using the R2 Platform to find genes that correlate with CBX family members’ expression (*p* < 0.01, FDR < 0.01). All data is freely available online, and the access is unrestricted and does not require patients’ consent or other permissions.

### 4.12. Statistical Analyses

Statistical analyses were carried out with GraphPad Prism 8.0 software (GraphPad Software, Inc., La Jolla, CA, USA). Multiple comparisons were performed with the ANOVA test. The correlation between two variables was assessed with Spearman’s rank correlation coefficient (r).

## 5. Conclusions

Our research uncovered a significant association between cancer stemness and an elevated expression of HP1 group of CBX family members, especially—CBX3, where the association was robust and universal regardless of the tumor type. On the other hand, the relation of Pc group of CBX members is rather tumor and protein specific, albeit for CBX7, is significantly negative in all tested tumor types. Both proteins are strictly associated with the epigenetic regulation of gene expression, although interacting with distinct binding partners and hence harnessing diverse modes of action. Further studies are needed to determine whether epigenetic functions mediated by Cbx3/HP1γ and Cbx7 are sufficient to promote cancer stemness. Additionally, targeting the epigenetic machinery has been proven as an efficient anti-cancer strategy in several tumor types, albeit it remains an open question whether Cbx3/HP1γ or Cbx7 epigenetic factors could become novel therapeutic targets in stem cell-like tumors.

## Figures and Tables

**Figure 1 ijms-23-13083-f001:**
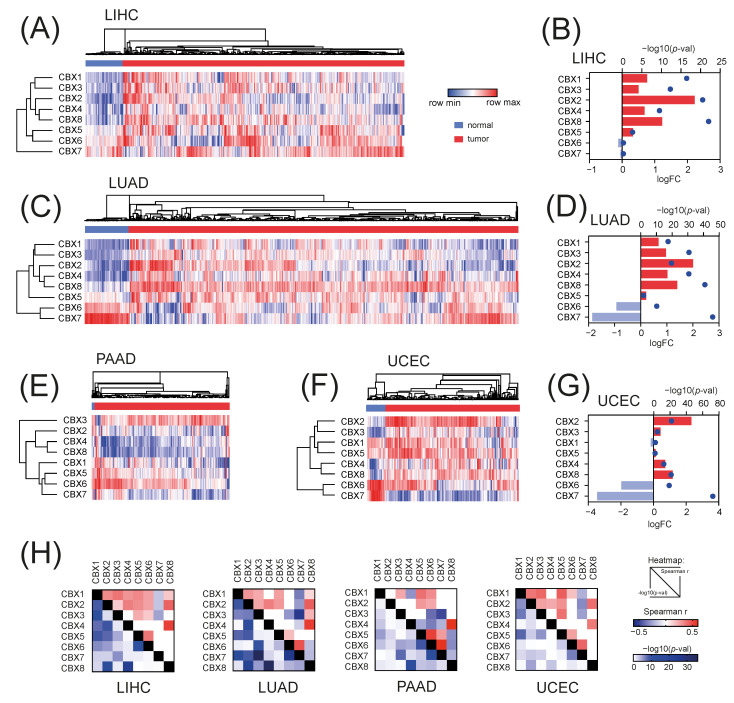
The expression of CBX family members in selected TCGA tumors. The level of CBX family members and differential CBX expression between tumor and normal adjacent tissues in (**A**,**B**) liver, (**C**,**D**) lung, (**E**) pancreatic and (**F**,**G**) uterine tumors. In the heatmaps, blue and red denotes low and high expression, respectively. Sample groups are color-coded accordingly: blue—normal tissue; red—tumor tissue. Bar plots depict the fold change (log2) of CBX expression in tumor vs. normal tissues. The statistical significance (−log10(*p*-value)) is marked with blue dots. (**H**) The correlation of CBXs expression in selected TCGA tumors. The upper part of the heatmaps presents the Spearman correlation coefficient, while the lower part shows the statistical significance of the correlation (−log10(*p*-value)).

**Figure 2 ijms-23-13083-f002:**
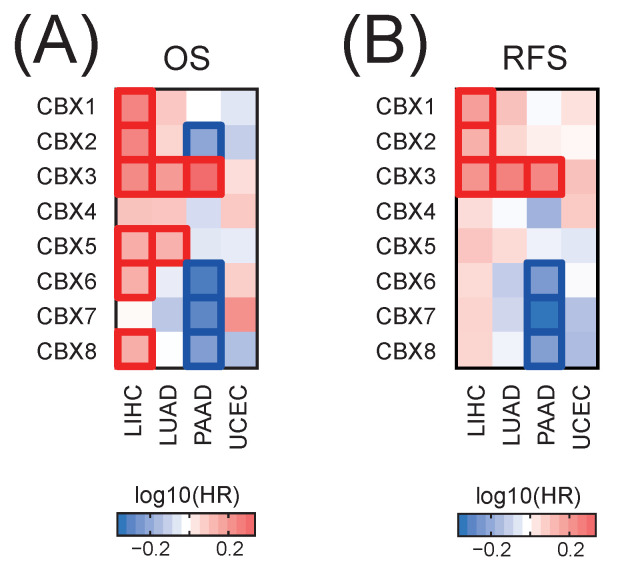
The association between CBX family members and patient prognosis in selected TCGA tumors. The heatmap of the hazard ratio of death (**A**) or relapse (**B**) for patients with high expression of specific CBX family members (with the mean expression as a cut-off). Red and blue denote higher or lower hazard ratios, respectively. Bordered squares denote statistically significant HRs (*p* < 0.05).

**Figure 3 ijms-23-13083-f003:**
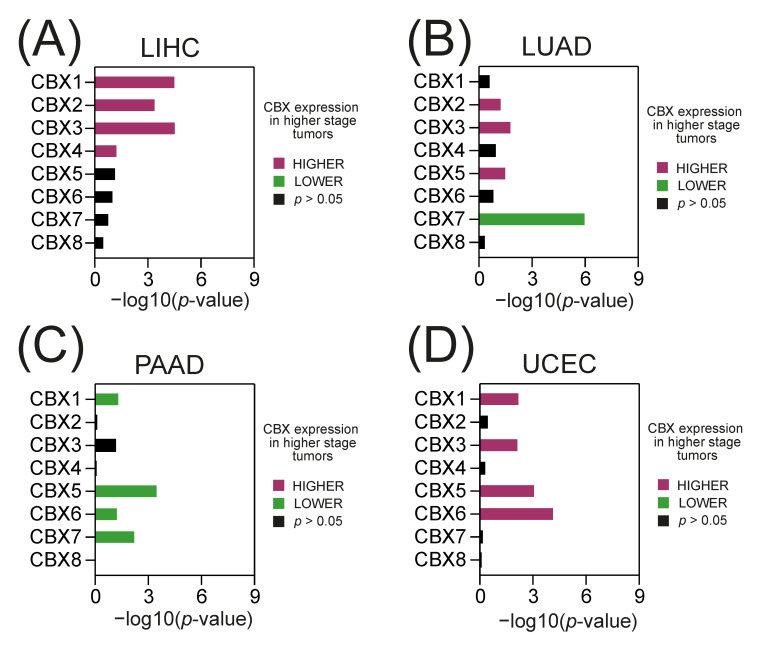
The association between CBX family members and disease stage in selected TCGA tumors. The association between CBXs expression and tumor stage—either lower (green) or higher stage (purple), as determined with the Spearman correlation test (−log10(*p*-value)) in (**A**) LIHC, (**B**) LUAD, (**C**) PAAD, and (**D**) UCEC data.

**Figure 4 ijms-23-13083-f004:**
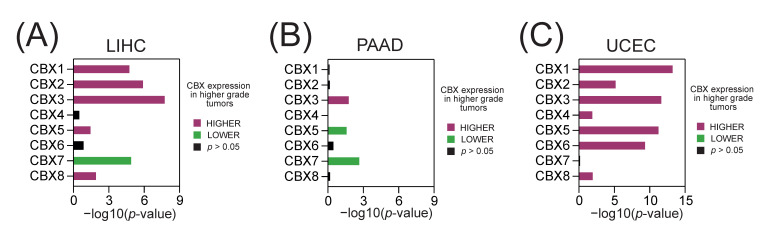
The association between CBX family members and tumor grade in selected TCGA tumors. The association between CBXs expression and tumor grade—either lower (green) or higher stage (purple), as determined with the Spearman correlation test (−log10(*p*-value)) in (**A**) LIHC, (**B**) PAAD, and (**C**) UCEC data.

**Figure 5 ijms-23-13083-f005:**
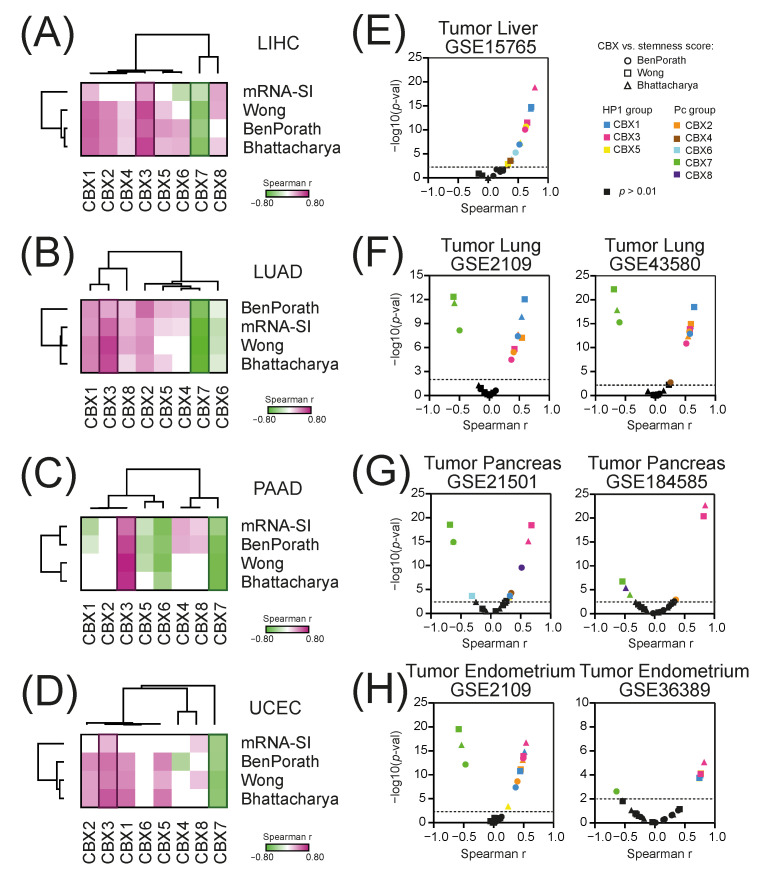
The correlation of CBX family members expression with cancer stemness. (**A**–**D**) The heatmaps of Spearman’s correlation between CBXs’ expression and four distinct stemness indices (mRNA-SI, Ben-Porath signature, Wong signature, Bhattacharya signature) in (**A**) LIHC, (**B**) LUAD, (**C**) PAAD, and (**D**) UCEC tumors. Purple and green denote either positive or negative correlation, respectively. Only statistically significant associations are shown (*p* < 0.05). (**E**–**H**) The correlation of CBX expression with cancer stemness scores in several additional datasets for (**E**) liver, (**F**) lung, (**G**) pancreatic, and (**H**) endometrial tumors. CBX members are color-coded as presented in the legend. Circles, squares, and triangles encode for Ben-Porath, Wong, and Bhattacharya signatures, respectively. Statistically insignificant correlations are colored in black.

**Figure 6 ijms-23-13083-f006:**
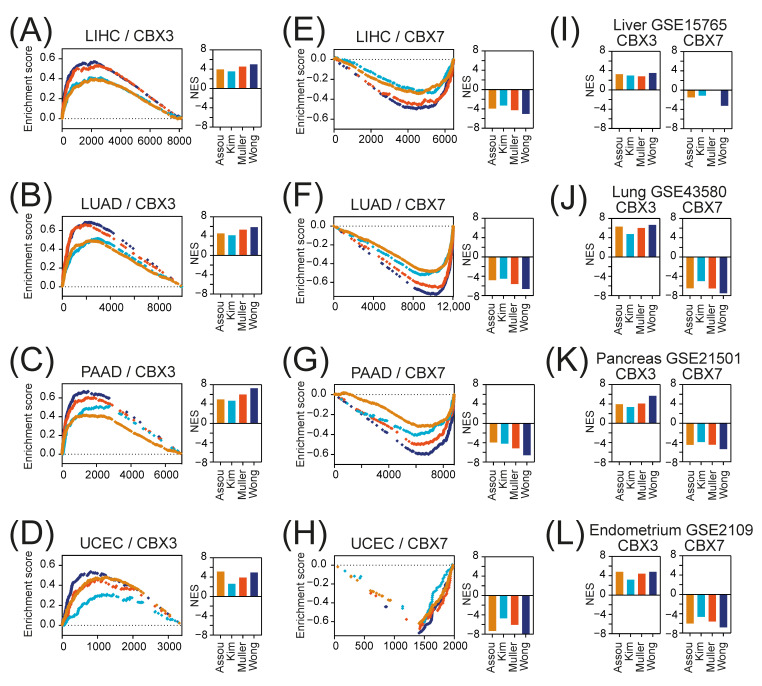
Stemness gene signatures in CBX3- and CBX7-associated transcriptome profiles. (**A**–**D**) The GSEA of all significantly correlated genes to CBX3 revealed significant enrichment (*p* < 0.0001) of stemness-associated gene signatures (Assou_ESC_, Kim_Myc, Muller_Plurinet, Wong_ESC_core) in (**A**) LIHC, (**B**) LUAD, (**C**) PAAD, and (**D**) UCEC. The normalized enrichment score (NES) for each gene signature is plotted in the bar graph. (**E**–**H**) Similarly, the GSEA of all significantly correlated genes to CBX7 revealed significant depletion (*p* < 0.0001) of stemness-associated gene signatures (Assou_ESC_, Kim_Myc, Muller_Plurinet, Wong_ESC_core) in (**E**) LIHC, (**F**) LUAD, (**G**) PAAD, and (**H**) UCEC. The normalized enrichment score (NES) for each gene signature is plotted in the bar graph. (**I**–**L**) Accordingly, the GSEA of all significantly correlated genes to CBX3 (left panel) or CBX7 (right panel) revealed significant enrichment or depletion of stemness-associated gene signatures, respectively, in (**I**) liver GSE15765, (**J**) lung GSE43580, (**K**) pancreatic GSE21501, and (**L**) endometrial carcinoma GSE2109.

**Figure 7 ijms-23-13083-f007:**
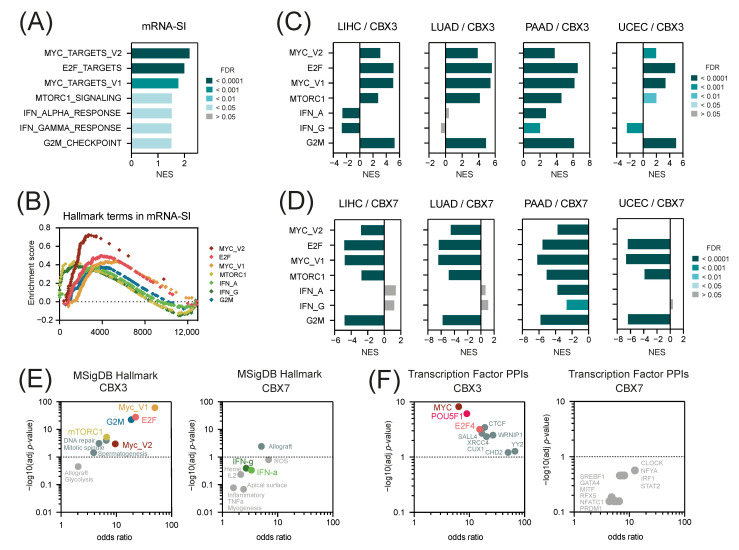
The transcriptome profiles associated with the expression of CBX3 are enriched with c-Myc and E2F transcription factor targets that significantly mirror the enrichment of mRNA-SI gene signature with the “Hallmarks of cancer” terms. (**A**,**B**) The enrichment of mRNA-SI gene signature with the „Hallmarks of cancer” terms determined using the GSEA analysis with the MSigDB Hallmark (v7.4) collection as a reference. Bars—Normalized Enrichment Score (NES). Bars are color-coded according to the statistical significance (FDR) as shown in the legend. (**C**,**D**) The CBX3-associated transcriptome profiles are enriched (**C**) while the CBX7-associated transcriptome profiles are depleted (**D**) with the “Hallmark of cancer” terms specific for cancer stem cell-like tumors. Bars—Normalized Enrichment Score (NES). Bars are color-coded according to the statistical significance (FDR) as shown in the legend. (**E**) The Enrichr tool [34] confirmed significant enrichment of CBX3-related gene expression profile with targets for c-Myc and E2F transcription factors. The top 100 most relevant genes for CBX3 (**left** panel) and CBX7 (**right** panel) determined with ARCHS4 RNA-seq gene-gene co-expression matrix were used as inputs, and MSigDB Hallmark collection was used as a reference. (**F**) Additionally, the Enrichr tool revealed significant enrichment of CBX3-related gene expression profile with targets for Oct-3/4 transcription factor (encoded with POU5F1 gene).

**Figure 8 ijms-23-13083-f008:**
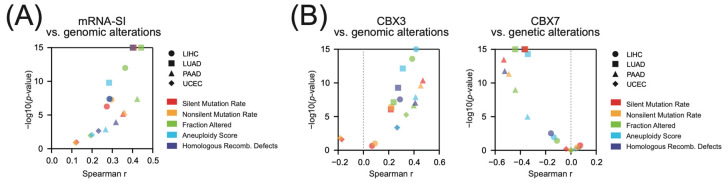
The association between CBX3 and CBX7 expression and genomic alterations in selected TCGA tumors. (**A**) Cancer stemness positively correlates with genomic alterations in LIHC (circle), LUAD (square), PAAD (triangle), and UCEC (rhombus) tumors. Several distinct scores were used to determine the association between cancer stemness (mRNA-SI) and distinct genomic alterations: silent (red) or non-silent mutation rate (yellow), the fraction of altered genome (green), aneuploidy score (light blue), homologous recombination defects score (purple) in selected cancer types. (**B**) The expression of CBX3 positively, while CBX7 negatively correlates with genome alterations in tested tumors. Tumor types are coded with figures accordingly: circle—LIHC, square—LUAD, triangle—PAAD, rhombus—UCEC, while genomic aberrations are color-coded as follows: red—silent mutation rate, yellow—non-silent mutation rate, green—the fraction of altered genome, light blue—aneuploidy score, purple—homologous recombination defects score.

## Data Availability

The datasets supporting the conclusions of this article are available in the TCGA and GEO repositories (TCGA: LIHC, LUAD, PAAD, and UCEC datasets (Firehose Legacy) from the cBioportal, www.cBioportal.org; GEO: GSE15765, GSE2109, GSE43580, GSE21501, GSE184585, GSE36389 from the R2 platform, http://r2.amc.nl, accessed on 5 January 2022).

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
