# Peer review of "Mining Transcriptomic Data to Uncover the Association between CBX Family Members and Cancer Stemness"

_ijms, 2022, doi:10.3390/ijms232113083_

Round 1

Reviewer 1 Report

The manuscript "Mining transcriptomic data to uncover the association between CBX family members and cancer stemness" reports a detailed multi-tool and all-round analysis of the public databases on gene expression levels and mutations for the entire family of CBX - a part of a histone methylation readers complex PRC1. The authors conclude that there are important changes, both upregulation and downregulation of different members of CBX family which appear to be individual for different cancers analyzed in the study. Moreover, CBX members are differentially associated with the cancer stem cell signatures and mutation burden. The reported results are overall novel and interesting, and, despite the lack of any experimental validation this work is certainly a thorough and significant for further understanding of the roles of CBX in cancers.

The work is overall well written and organized, however, certain concerns must be addressed by the authors:

- It is  unexpected, that the data from Oncomine is presented as the table. It would be more useful to visualize these data and put it as a supplementary figure. 

- Panels A, C, E and F are complicated but serve no purpose, they are not even discussed in the text. I suggest they should be moved in the supplement. In contrast, a more important quantification of PAAD cohort is absent. 

- The presentation of the p value as bars on Figure 3 and Figure S2 is confusing. Why not use the same format as on Figure 1 B,D,G?

- it would be interesting to analyze the expression levels of other components of PRC1 complex and their correlation with CBX homologs. 

Author Response

Point-by-point answers to the Reviewer's criticisms.

We thank the Reviewer for his/her time and criticisms. It has certainly helped us improve the quality and focus of our manuscript. All the changes made to the manuscript are highlighted in yellow. 

  1. It is  unexpected, that the data from Oncomine is presented as the table. It would be more useful to visualize these data and put it as a supplementary figure. 

We decided to present the data from the Oncomine database in a tabular form to show the exact differences in CBX family members’ expression between tumor and normal adjacent tissue (across four distinct tumor types: lung, liver, pancreatic, and uterine tumors), not just the number of Oncomine datasets that fulfills the criteria of statistically significant differences between two compared group of samples. In Table S1, we demonstrate the fold change of CBX expression in tumor vs. normal adjacent tissues, followed by the statistical significance of differential expression across Oncomine datasets encompassing lung, liver, pancreatic, and uterine tumors. In our opinion, this is a clear and informative way to present this type of data.

  1. Panels A, C, E and F are complicated but serve no purpose, they are not even discussed in the text. I suggest they should be moved in the supplement. In contrast, a more important quantification of PAAD cohort is absent. 

In our manuscript, we decided to present the expression of CBX family members in selected tumor types (LIHC, LUAD, PAAD, UCEC) in two different ways: (i) in each sample individually - using the heatmaps (Figure 1A, C, E, and F) and (ii) as a fold change between tumor and normal adjacent tissue - with the bar plots (Figure 1B, D, G). The heatmaps are an excellent way to present the heterogeneity of expression of distinct CBX family members across tumor samples within one tumor type. On the other hand, bar plots are more suitable to explain the difference between the mean expression of the analyzed gene in tested groups. Therefore, we believe that both heatmaps and bar plots should be used to visualize the CBX family gene expression pattern in analyzed tumor types.

In the TCGA PAAD cohort, there are only 4 samples collected from the normal adjacent tissue and 178 samples collected from tumor tissue. Therefore, we could not calculate the exact fold change of CBX’s expression in this tumor type. However, for selected CBX members, namely CBX1, CBX3, and CBX5, we have demonstrated significant upregulation in tumor tissues vs. normal tissues using the Oncomine datasets (Table S1).

  1. The presentation of the p value as bars on Figure 3 and Figure S2 is confusing. Why not use the same format as on Figure 1 B,D,G?

We decided to present the association between CBX family members’ expression and the tumor grade (Figure 3) or tumor stage (Figure 4) using the bar plots of statistical significance (-log10(p-value)) determined with the Spearman correlation test. In our opinion, the visualization of all comparisons, i.e., between each of four tumor grades for each of eight CBX members in each of the tested tumor types would be less clear. We believe that the presentation of the power of association (statistical significance) is the best way to demonstrate whether the expression of the tested gene correlates with tumor grade/stage.

  1. it would be interesting to analyze the expression levels of other components of PRC1 complex and their correlation with CBX homologs. 

In our work, we have determined the association between the CBX family members’ expression and several molecular traits of solid tumors, highlighting the association of CBX proteins with cancer stemness. CBX proteins are classified into two distinct groups: (i) CBX2/4/6/7/8 constitute the canonical components of polycomb repressive complex 1 (PRC1), while (ii) CBX1/3/5 (also known as HP1β, HP1γ, and HP1α, respectively) interact through their C-terminal chromoshadow (CDS) domain with a TRIM28 scaffold protein, facilitating chromatin compaction. We have discovered that cancer stemness is significantly positively associated with an elevated expression of the HP1 group of CBX family members, especially the CBX3/HP1g exhibiting a robust and universal positive association with tumor dedifferentiation status regardless of the tumor type. On the other hand, the Polycomb group of CBX members demonstrated rather a tumor and protein specific association with cancer stemness, albeit for CBX7 we observed a consistent negative association in all tested tumor types. Both groups of CBX proteins are strictly associated with the epigenetic regulation of gene expression, although they interact with distinct binding partners and hence harness diverse modes of action. Our current work is focused on determining the exact molecular mechanism of CBX-related cancer stemness acquisition, including (i) the analysis of other components of the Polycomb Repressive Complex as well as (ii) the analysis of TRIM28-HP1 interacting proteins. However, the primary goal of this manuscript was to outline the vast divergence of associations between the CBX family members and cancer dedifferentiation status.

Reviewer 2 Report

In my opinion, the submitted manuscript „ Mining transcriptomic data to uncover the association between CBX family members and cancer stemness” meets aims and scope of „International Journal of Molecular Sciences” Section „ Molecular Oncology” and may be accepted after minor revision.

1.       In my opinion, the authors in the introduction to the publication, should explain in more detail how they analyzed data from the Internet (only in the abstract of the publication it was mentioned that bioinformatics tools were used, there is no such information in the introduction). The way of analyzing the data should be described in such a way that another independent team of scientists may repeat the experiment independently of the authors of the publication.

2.       In my opinion, the authors should complete their publication with information about whether it is known how to influence the expression of CBX proteins. Can it be considered that finding factors influencing this expression could be a means of anti-tumor activity? Is it known about such research?

3.       In the abbreviation „et al.” (from Latin: et alia) the dot after the word „al” is needed (e.g. line 201), it  also could be written in italics (line 255, 366, 393,417) .

4.       When using the abbreviation „CBX7”, you sometimes write it in the form of „Cbx7”, and sometimes: „CBX7” – write consistently in the chosen way.

5.       There is no abbreviation list in the publication – please complete it.

Author Response

  1. In my opinion, the authors in the introduction to the publication, should explain in more detail how they analyzed data from the Internet (only in the abstract of the publication it was mentioned that bioinformatics tools were used, there is no such information in the introduction). The way of analyzing the data should be described in such a way that another independent team of scientists may repeat the experiment independently of the authors of the publication.

According to the Reviewer’s suggestion, we have edited the Introduction section of our manuscript to highlight all the bioinformatic tools or platforms that were used in our analyzes. Specifically, the current version is as follows:

[…]

“Here, we have exploited publicly available transcriptomic data from 4 distinct types of solid tumors, namely liver, lung, pancreatic, and uterine carcinoma, to delineate the association between CBX family members and cancer stemness measured with previously reported stemness scores [3–6]. Firstly, we analyzed the expression pattern of all CBX family members in tumor vs. normal adjacent tissues using publicly available TCGA [27], Oncomine [28], and CPTAC data [29] from the cBioportal, Oncomine, and UALCAN databases, respectively. Using the GSCA platform, we determined the frequency of alterations in CBX family members. According to the GEPIA2 database [30], for several CBX members, we reported a significant association with TCGA cancer patients’ survival, mostly with a worse outcome. Furthermore, the correlation with clinicopathologic features revealed significant upregulation of CBX3 followed by significant downregulation of CBX7 in more advanced disease and in higher-grade tumors, as determined with the TISIDB database.

Using a transcriptome-based stemness index (mRNA-SI) and other stem cell-derived gene expression signatures [3–6], we analyzed the association between CBX family members and cancer dedifferentiation status. We observed that regardless of the TCGA tumor type or the stemness score, the expression of CBX3 is significantly positively and the level of CBX7 is significantly negatively associated with cancer dedifferentiation, which was further validated with additional datasets from the R2 database.

Furthermore, the Gene Set Enrichment Analysis (GSEA) [31] revealed that transcriptome profiles of CBX3HIGH tumors are significantly enriched, while CBX7HIGH tumors are significantly depleted with stemness markers, respectively, further supporting our first observation. We further confirmed a robust enrichment of CBX3-associated transcriptome profiles with several “hallmarks of cancer” previously identified as upregulated in stemness-high tumors [3]. The opposite results were observed in CBX7HIGH tumors. Moreover, as determined with the Enrichr analysis tool, the CBX3-associated transcription factor protein-protein interaction network is significantly enriched with c-Myc and Oct-3/4 transcription factors, which are known pluripotency markers.” […]

All necessary details regarding the data analyzes with bioinformatic tools or platforms are described in the Material and methods section (4. Materials and methods; pages 12-14, lines 427-504). 

  1. In my opinion, the authors should complete their publication with information about whether it is known how to influence the expression of CBX proteins. Can it be considered that finding factors influencing this expression could be a means of anti-tumor activity? Is it known about such research?

According to the Reviewer suggestions, we have added a short comment to the Discussion section to indicate the druggable potential of CBX proteins and to imply the necessity for searching for chemical compounds targeting CBX family members. We have also emphasized the need for further studies to determine whether inhibiting CBX proteins would affect cancer stemness. The last part of the Discussion section is as follow (page 12, lines 430-439):

“Nowadays, it is well-established that the epigenetic dysregulation accelerates tu-morigenesis, and at least partially, facilitates cancer stemness. Several reports suggest, that CBX family members possess the potential to become druggable targets in distinct tumor types, including breast, lung, ovarian, esophageal, and colon cancer [11,66-69]. However, these reports do not raise the question of therapeutic targeting of the stem cell-like phenotype of solid tumors and do not prove the engagement of CBX family members in molecular mechanism modulating cancer stemness. Albeit several molecular inhibitors of CBX proteins are already under the development [70-72], further studies are necessary to determine whether they provide advantages in the treatment of de-differentiated tumors that strongly exhibit cancer stem cell-like characteristics.”

Also, we have concluded that the possibility to target therapeutically the members of CBX family is still an open question. (Conclusions, page 14, lines 527-530).

  1. In the abbreviation „et al.” (from Latin: et alia) the dot after the word „al” is needed (e.g. line 201), it  also could be written in italics (line 255, 366, 393,417) .

      We truly apologize for this error. It has been corrected in the current version of our manuscript (page 6, line 201; page 8, line 255; page 11, line 366; page 11, line 393; page 12, line 417).

  1. When using the abbreviation „CBX7”, you sometimes write it in the form of „Cbx7”, and sometimes: „CBX7” – write consistently in the chosen way.

      The name CBX7 (written in uppercase letters) denotes the name of the gene encoding the Cbx7 protein (written with its first letter as a capital letter). Therefore, when we describe the gene, we use all capital letters, while for the protein – only the first is capitalized. We have corrected the spelling across the whole manuscript.

  1. There is no abbreviation list in the publication – please complete it.

According to the IJMS instructions for the authors, there is no need to prepare a separate section for abbreviations. Although, as suggested by the Reviewer, we have included the list of abbreviations at the end of our manuscript.
